# Strength–Toughness of a Low-Alloy 0.25C Steel Treated by Q&P Processing

**DOI:** 10.3390/ma16103851

**Published:** 2023-05-19

**Authors:** Evgeniy Tkachev, Sergey Borisov, Yuliya Borisova, Tatiana Kniaziuk, Sergey Gaidar, Rustam Kaibyshev

**Affiliations:** 1Laboratory of Mechanical Properties of Nanostructured Materials and Superalloys, Belgorod State University, 308015 Belgorod, Russia; tkachev_e@bsu.edu.ru (E.T.);; 2Laboratory of Advanced Steels for Agricultural Machinery, Russian State Agrarian University—Moscow Timiryazev Agricultural Academy, 127550 Moscow, Russia; 3National Research Center “Kurchatov Institute”, Central Research Institute of Structural Materials “Prometey”, 191015 St. Petersburg, Russia

**Keywords:** Q&P process, microstructure, mechanical characterization, bainitic transformation, retained austenite, impact toughness

## Abstract

Quenching and partitioning (Q&P) treatments were applied to 0.25C steel to produce the microstructures that exhibit an improved balance of mechanical properties. The simultaneous bainitic transformation and carbon enrichment of retained austenite (RA) during the partitioning stage at 350 °C result in the coexistence of RA islands with irregular shapes embedded in bainitic ferrite and film-like RA in the martensitic matrix. The decomposition of coarse RA islands and the tempering of primary martensite during partitioning is accompanied by a decrease in the dislocation density and the precipitation/growth of η-carbide in the lath interiors of primary martensite. The best combinations of a yield strength above 1200 MPa and an impact toughness of about 100 J were obtained in the steel samples quenched to 210–230 °C and subjected to partitioning at 350 °C for 100–600 s. A detailed analysis of the microstructures and the mechanical properties of the steel subjected to Q&P, water quenching, and isothermal treatment revealed that the ideal strength–toughness combinations could be attributed to the mixture of the tempered lath martensite with finely dispersed and stabilized RA and the particles of η-carbide located in the lath interiors.

## 1. Introduction

Low- and medium-carbon advanced high-strength steels (AHSS), which have an improved combination of strength, ductility, and toughness, are inexpensive and promising materials for application in many industrial fields. To achieve the improved combination of mechanical properties, these steels are often subjected to a three-step quenching and partitioning (Q&P) heat treatment, which usually results in a mixed microstructure of tempered martensite (TM), retained austenite (RA), and bainite [1,2,3,4,5]. This heat treatment includes full or partial austenitization (I); quenching to temperature (QT) between the martensite-start (Ms) and martensite-finish (Mf) temperatures to obtain an incomplete martensitic transformation (II); and partitioning at temperature PT ≥ QT with a final cooling to room temperature (III). The increased fraction of RA in Q&P steels compared to those produced after conventional quenching and tempering (Q&T) treatments is attributed to the redistribution of carbon from the supersaturated martensite to the untransformed austenite during the quenching and partitioning stages [1,6,7]. Depending on the overall carbon content, the quenching and partitioning temperatures, and the partitioning time (Pt), different fractions of stabilized austenite can be obtained [8,9,10]. If some fraction of RA is not sufficiently stabilized during the partitioning stage, it will transform into martensite during the final quenching to room temperature. This martensite is referred to as an untempered martensite (UM).

While the Q&P process is widely used to improve the plasticity of cold-rolled sheet steels for use in the automotive industry [11,12,13,14,15], the relationship between the strength and the impact toughness in thick-plate Q&P steels with multiphase microstructures is still debated. Recent studies suggest that the steel microstructures composed of carbon-depleted martensite and an increased fraction of RA are generally desired to accommodate significant plastic strain during tensile deformation due to the transformation-induced plasticity (TRIP) effect [14,15,16]. On the other hand, although the coarse blocky austenite significantly improves the steel’s ductility, it was shown to have detrimental effects on the impact toughness [17,18]. This may be due to an increase in the effective grain size controlling the toughness and lower stability of RA, which transforms to brittle secondary martensite during the final cooling [15,19]. Bagliani et al. argued that the presence of UM in the microstructure can have a prominent effect on the material’s mechanical behavior, and especially on toughness [20]. Large UM–A constituents cannot effectively inhibit crack propagation and result in low fracture toughness [20,21,22]. To improve toughness, the bainitic transformation of untransformed islands of RA during partitioning can be utilized. The finely dispersed mixture of lath bainite ferrite, lath martensite, and film-like retained austenite can effectively inhibit crack propagation [21,22]. Huang et al. showed that the microstructure composed of martensite and bainite obtained during uncompleted isothermal transformation, followed by Q&P, can improve the material’s impact toughness without significant losses in strength [17]. Thus, the strength–toughness relationship in Q&P steels deserves more detailed investigation. The evolution of retained austenite during bainite transformation in the partitioning stage and its effect on impact behavior are of particular interest.

Another important issue in the context of improving the strength and toughness of Q&P steels is the formation of carbide particles. The numerous plate-shaped cementite particles formed along the lath boundaries decrease the amount of carbon available for the enrichment of RA and are unfavorable for the steel’s toughness; therefore, they should be avoided [23,24,25]. In practice, the precipitation of cementite is usually suppressed by alloying the steel with ~1.5 wt.% silicon. However, alloying with Si does not inhibit the formation of transition η-carbide during martensite tempering. Hence, the precipitation of transition carbides inside the martensite laths is often observed after partitioning at PTs of 300–400 °C [5,6,26]. The effect of transition carbides on the toughness of low-alloy steels is not clear. The precipitation of η-carbide particles with a length of ~100 nm and a width of ~15 nm is accompanied by a slight increase in toughness, whereas the coarse plates of this carbide that are formed after tempering at 400 °C lead to the pronounced temper embrittlement of low-alloyed Q&T steel [27]. Kaar et al. showed that carbon partitioning to austenite competes with the trapping of carbon at dislocations, grain boundaries, and carbides in martensite and bainitic ferrite in 0.2C steel with 1.5 wt.% Si [28]. Thus, the precipitation of carbide particles during Q&P and its effect on the material’s mechanical properties should be also considered.

In the present study, various Q&P processes were carried out to produce multiphase microstructures containing martensite, bainite, and film-like RA. These microstructures were studied in detail and compared with those obtained after water quenching and isothermal treatment at 350 °C. We demonstrated that ~85% tempered martensite and ~15% bainitic ferrite mixed with RA obtained by Q&P results in an excellent combination of strength and toughness. Furthermore, the kinetics of austenite decomposition during bainitic transformation and the tempering of primary martensite during the partitioning stage and its effect on the material’s mechanical properties are discussed.

## 2. Materials and Methods

A 50 kg ingot of the 0.25 wt.%C low-alloy steel (chemical composition of 0.25 wt.%C, 1.47 wt.% Mn, 1.60 wt.% Si, 0.51 wt.% Cr, 0.217 wt.% Mo, ≤0.005 wt.% S, ≤0.01 wt.% P) was produced by air induction melting followed by electroslag remelting. The amounts of N and O in the steel were determined as 0.01 and 0.002 wt.%, respectively, using the gas fusion technique. The increased Si content of 1.6 wt.% in the studied steel was shown to be an effective inhibitor for Fe_3_C precipitation at tempering temperatures ≤400 °C [27]. The steel ingot was homogenized at 1150 °C for 4 h and hot forged into a slab with sectional dimensions of 150 × 70 mm, followed by air cooling. Then, a plate with a thickness of 45 mm was cut from the slab. The steel plate was reheated to 1000 °C for 2 h and then hot-rolled to 12 mm thickness (70% total reduction) through 6 passes in the temperature range from 900 °C to 850 °C, followed by air cooling. To maintain essentially isothermal rolling conditions, the plate was reheated at 900 °C for 5 min prior to each pass. A 2-roll mill with a maximum rolling force of 2500 kN was used.

In order to obtain the desired microstructures of tempered martensite with a significant fraction of stabilized RA by Q&P processing, the proper selection of the quenching and partitioning temperatures should be carried out. The maximum volume fraction of RA can be estimated using the method originally proposed by Speer et al. [6,29]. This approach is based on ideal constrained carbon equilibrium (CCE) conditions and incorporates the following simplifying assumptions: the carbon atoms fully partition into austenite (1), the α–γ interface is immobile (2), and neither austenite decomposition nor carbide precipitation occurs during partitioning (3). As a function of quenching temperature, the martensite fraction, $f_{M}$, can be estimated by the Koistinen–Marburger (K–M) relationship [30]:(1)$$f_{M}=1-e^{-1.1\cdot 10^{-2}\left(M_{s}-\mathrm{QT}\right)}$$
where QT is the quenching temperature. The *Ms* temperature can be estimated using the empirical equation that incorporates the effect of carbon and the alloying elements [31]:(2)$$M_{S}=539-423C-30.4\mathrm{Mn}-7.5\mathrm{Si}-12.1\mathrm{Cr}-7.5\mathrm{Mo}$$
where C, Mn, Si, Cr, and Mo are the wt.% of carbon, manganese, silicon, chromium, and molybdenum, respectively. The phase fractions of initial martensite, retained austenite, and secondary martensite were predicted as functions of QT using Equations (1) and (2) (Figure 1).

The maximum possible fraction of RA of about 23% corresponds to QT = 233 °C, and the formation of the secondary martensite is not predicted below this temperature. Thus, 2 different temperatures below the optimum predicted QT of 230 °C and 210 °C were selected for investigation. The selected PT of 350 °C is slightly below the *Ms* point and is considered the optimum temperature to ensure the partitioning of carbon to austenite; it also prevents the pronounced growth of carbide particles, according to the previous results for martensite tempering in the studied steel [24]. In addition to series of Q&P treatments, the samples subjected to water quenching (As-quenched) and isothermal treatments (IT) were studied to comparatively analyze the microstructure and mechanical properties (Figure 2a). The cooling rate of the steel samples in the salt bath is about 50 °C/s, which is well above the critical cooling rate and is thus sufficient to form the initial martensite during the first step of Q&P processing [20]. The sample’s surface after the heat treatment was mechanically ground, removing ~0.1 mm, to eliminate the issues related to the decarburization layer.

A high-precision Bahr DIL 805 A/D dilatometer was used to measure the length changes during the Q&P process and fast cooling to room temperature. X-ray diffraction (XRD) experiments were performed using a Rigaku Ultima IV diffractometer with CuKα1 radiation for the 2θ range of 35–105° at a scanning rate of 1 degree/min. The volume fraction of the retained austenite was calculated using the direct comparison method by comparing the integrated intensities of austenite and the martensite [32]:(3)$$V_{\gamma }=\frac{\frac{1}{n}\sum_{i=1}^{n}\frac{I_{l}^{\left(\gamma \right)}}{R_{l}^{\left(\gamma \right)}}}{\frac{1}{m}\sum_{i=1}^{n}\frac{I_{l}^{\left(\alpha^{\prime }\right)}}{R_{l}^{\left(\alpha^{\prime }\right)}}+\frac{1}{n}\sum_{i=1}^{n}\frac{I_{l}^{\left(\gamma \right)}}{R_{l}^{\left(\gamma \right)}}},$$
where *n* and *m* are the number of visible diffraction peaks for each phase, and *R_l_* is the theoretical intensity values. The average concentration of carbon in RA was calculated using following relationship [33]:(4)$$C_{\gamma }=\frac{a_{\gamma }-3.547}{0.046},$$
where *C_γ_* is the carbon concentration in RA in wt.%. a_γ_ is the lattice parameter of austenite estimated on the (200) *γ*, (220) *γ*, and (311) *γ* peaks as:(5)$$a_{\gamma }=\frac{\lambda \sqrt{h^{2}k^{2}l^{2}}}{2\sin\theta },$$
where *λ* is the wavelength, *h*, *k*, and *l* are the Miller indices of a plane, and *θ* is the Bragg angle.

The dislocation densities in martensite were estimated using the modified Williamson–Hall method by measuring full width at half maximum (FWHM) values for the (110) *α*’, (200) *α*’, and (211) *α*’ peaks and using the following simplified equation [34,35]:(6)$$\Delta K=\frac{\alpha }{D}+\left(\frac{\pi Mb^{2}}{2}\right)\rho_{\alpha }^{1/2}K^{2}\bar{C},$$
where Δ*K*
$=\left(\mathrm{FWHM}\left(\theta \right)\cdot 2\cos\left(\theta \right)\right)/\lambda$, *K* is the scattering vector defined by $K=2\sin\left(\theta \right)/\lambda$, *D* is the so-called apparent size parameter, *α* is a constant of 0.9, *M* is a parameter related to the outer cut-off radius of the dislocations (2.2 according to Ref. [35]), $\bar{C}$ is the average contrast factor of dislocations for a particular {hkl} reflection [36], and *b* is the Burgers vector.

Scanning electron microscopy (SEM) imaging and electron back-scattering diffraction (EBSD) analysis were performed using a FEI Quanta 600 FEG electron microscope operated at 20 kV. EBSD mapping of the samples was performed over the scan area of 100 × 100 μm using a 150 nm step size and analyzed using TSL-EDX OIM software and the open-source crystallographic toolbox MTEX 5.8 with the ORTools suite. The variant analysis was performed using the parent grain reconstruction method proposed by Niessen et al. [37] and assuming a Kurdjumov–Sachs (K–S) orientation relationship (OR) between martensite blocks inside individual prior austenite grains (PAG):

${\left\{111\right\}}_{\gamma }\|{\left\{011\right\}}_{\alpha },\;{\langle \bar{1}01\rangle }_{\gamma }\|{\langle \bar{1}\bar{1}1\rangle }_{\alpha }$ K-S OR.

Transmission electron microscopy (JEOL JEM 2100) was used to characterize the phases and precipitates in the selected steel samples. The samples for SEM observations and thin foils for TEM analysis were prepared using electrolytic polishing and twin-jet electropolishing techniques, respectively, using an electrolyte composed of 10% HClO_4_ solution and 90% CH_3_COOH. The dislocation density of the steel samples was also calculated from the EBSD data using point-to-point kernel misorientation analysis [38]:(7)$$\rho_{KAM}=\frac{2\cdot \theta_{KAM}}{b\cdot h},$$
where *h* is the scan step and $\theta_{KAM}$ is the average misorientation between a central point and its nearest neighbors, excluding the misorientations larger than 5°. Carbon concentration profiles were predicted using diffusion simulations in the DICTRA diffusion module (v26) with the MOBFE2 mobility database for ferrous alloys.

The flat specimens with a gauge length of 35 mm, a width of 7 mm, and a thickness of 3 mm were prepared for the tensile tests (Figure 2b). The tensile tests were performed at room temperature using an Instron 5882 testing machine at a constant loading rate of 1 mm/min. The elongation and reduction of the cross-section area of the tensile specimens were recorded using a non-contact digital image correlation (DIC) measurement system.

Rockwell C-scale measurements were used to characterize the hardness of the steel samples. Impact tests were performed with standard Charpy V-notch specimens (10 mm × 10 mm × 55 mm) on an Instron IMP460 impact testing machine equipped with an Instron Dynatup Impulse data acquisition system.

## 3. Results

### 3.1. Phase Analysis and Microstructure

The relative change in length (∆L/L_0_) during the partitioning treatment, including the austenitization at 900 °C (Stage 1), quenching at 210 °C (Stage 2), and partitioning at 350 °C for 600 s with the final cooling to room temperature (Stage 3) is shown in Figure 3a. The applied quenching temperatures of 210 °C and 230 °C are well below the *Ms* point (the measured *Ms* temperature is about 360 °C). As the length increase caused by the carbon partitioning to austenite is too small and undetectable by dilatometry [39,40], the observed specimen expansion during isothermal holding at 350 °C can be directly attributed to the austenite decomposition during the bainitic transformation [39,41,42]. It should also be noted that the absence of a length increase during the final quenching to room temperature implies that no secondary martensite was formed upon cooling to room temperature.

The measured volume fractions of RA determined using the XRD and magnetic saturation methods are given in Table 1. The RA fraction in the as-quenched sample is too small to be detected by the X-ray method and was therefore measured only by the magnetic saturation method. The volume fractions of primary martensite after various Q&P routes were obtained by applying a simple lever rule to the dilatometric curve, as illustrated in Figure 3b. Then, the fractions of bainitic ferrite were calculated, assuming that M_INITIAL_ + RA_FINAL_ + BF = 1. According to these results, the volume fractions of the retained austenite prior to partitioning are 17% and 15% at 230 and 210, respectively. A decrease in the quenching temperature from 230 °C to 210 °C has no significant effect on the final fraction of RA in the steel samples subjected to the partitioning treatment. In contrast, the volume fraction of RA decreases markedly with the increasing the isothermal holding time at 350 °C.

The overall carbon content in α-Fe and carbides, $X_{C}^{\alpha +carbides}$, including bainitic ferrite, tempered martensite, and carbides, can be calculated, taking into account the volume fractions of RA, $f_{C}^{\gamma }$*,* and α-Fe, $f_{C}^{\alpha }=1-f_{C}^{\gamma }$*,* using the mass balance equation for carbon:(8)$$f_{C}^{\alpha }X_{C}^{\alpha +carbides}+f_{C}^{\gamma }X_{C}^{\gamma }=X_{C}^{0}\;,$$
where $X_{C}^{\alpha +carbides}$ and $X_{C}^{\gamma }$ are the weight fractions of carbon in α-Fe and RA, respectively, and $X_{C}^{0}$ is the nominal carbon concentration. The obtained values of $X_{C}^{\alpha +carbides}$ increase from 0.15 wt.% to 0.20 wt.% for QTs of both 210 °C and 230 °C with the Pt increasing from 20 s to 600 s. The decomposition of RA during the partitioning stage is accompanied by an increase in the volume fraction of bainitic ferrite (Table 1). Almost the same RA fractions are observed in the QP samples subjected to partitioning at 350 °C for 600 s and in the sample that was austempered at 350 °C for 600 s.

Figure 4 shows the SEM micrographs of the steel samples after different Q&P treatments. The observed microstructures mostly consist of martensite/austenite (M/A) constituents, with a notable fraction of bainitic ferrite (BF) and retained austenite (RA). Two distinct morphologies of RA can be distinguished. The retained austenite with a film-like morphology was observed in a martensitic matrix, while islands of RA with irregular shapes are located within the bainitic ferrite.

The microstructures of the Q&P samples obtained by EBSD are shown in Figure 5, and the microstructural parameters are given in Table 2. The average block width in the Q&P samples quenched at 210 °C is somewhat lower than that in the samples quenched at 230 °C. An increase in the Pt does not affect the block width, suggesting that the blocks of initial martensite and bainitic ferrite are characterized by similar sizes. The islands of RA in the Q&P samples are mainly observed along the sub-block boundaries of the bainitic ferrite. An increase in the volume fraction of bainitic ferrite during isothermal holding at 350 °C indicates that the carbon enrichment in austenite can be realized during carbon partitioning from both martensite and bainite to austenite. Nevertheless, the measured carbon content in RA was about 1 wt.% in all studied Q&P samples.

Figure 6 presents the TEM microstructure of a Q&P-processed sample (QT = 230 °C Pt = 20 s), which consists of lath martensite with film-like RA located at the lath boundaries and bainite with RA islands. The carbide particles with a plate-like shape precipitated after a short partitioning time are often arranged within the large laths of primary martensite (Figure 6c). These particles were identified as orthorhombic η-carbide (Fe_2_C) using selected area diffraction (Figure 6e). The η-carbide is formed during tempering at 200–400 °C in the studied steel [27]. By increasing the holding time from 20 s to 600 s, the length of the carbide particles increases significantly (Figure 7, Table 2).

The as-quenched martensite and the martensite subjected to a short partitioning treatment for 20 s are characterized by relatively high dislocation densities, as revealed by both the modified Williamson–Hall method (*ρ* _XRD_) and the KAM analysis (*ρ*
_KAM_). An increase in Pt from 20 s to 600 s leads to a moderate decrease in the dislocation density in the Q&P samples.

The SEM and TEM images of the microstructure of the steel sample after isothermal treatment at 350 °C are shown in Figure 8. Figure 8a shows that the RA in this sample has the same morphology as the RA in bainite in the Q&P samples. Moreover, the distinguishable carbide particles are not observed by SEM. Nevertheless, TEM reveals the precipitation of a small number of η-carbide particles inside the blocks with a high dislocation density (Figure 8b).

The diffraction spots highlighted by the red arrows in (Figure 8c) represent the Fe_3_O_4_ surface oxide, zone axis <111>, which was often reported to form epitaxially on the surface of the samples during storage; it has a (110)_α_//(111)Fe_3_O_4_, <113>_α_//<012>Fe_3_O_4_ orientation relationship with a ferrite matrix [43,44]. As the isothermal treatment was performed slightly below the *Ms* temperature, it is difficult to determine whether these particles were formed in blocks of bainitic ferrite or lath martensite. Meanwhile, the dimensions of the η-carbide particles in the A900-IT350 (600 s) sample do not differ significantly from those observed in tempered martensite in the Q&P samples.

### 3.2. Mechanical Properties

The stress–strain curves obtained during uniaxial tensile tests of the steel specimens subjected to different heat treatments are shown in Figure 9. Among the studied samples, the as-quenched steel exhibits the highest yield strength (YS) of 1360 MPa and an ultimate tensile strength (UTS) of 1740 MPa (Table 3).

The YS values for the Q&P specimens range between 1090 MPa and 1230 MPa, and the total elongation (TE) is 9.5% to 11.2%. The quenching temperature appears to have a weak effect on the tensile properties of the Q&P specimens. In contrast, a concurrent increase in YS and decrease in UTS correlates with an increase in the duration of the partitioning stage. The elongation to failure and uniform elongation of the Q&P specimens decrease progressively as the holding time at 350 °C increases, which can be attributed to the decomposition of RA during bainite transformation (Table 1). The work hardening rate decreases more rapidly in the samples treated for 20 s compared to the samples treated for 600 s. Isothermal treatment at 350 °C results in the lower YS values of 1070 MPa and UTS of 1340 MPa as compared to the Q&P samples. Nevertheless, the yielding behavior of the IT specimen at a true strain of ~0.005 is quite similar to that of the Q&P specimens treated for 20 s (Figure 9b).

CVN tests were carried out to determine the effect of the Q&P and isothermal treatments on the room-temperature toughness of the steel (Table 3). The Q&P process with a Pt of 20 s results in an impact toughness of ~40 J, which is somewhat lower than that of the as-quenched steel (~50 J). An increase in Pt from 20 s to 600 s is accompanied by more than a two-fold increase in the impact toughness. It should be noted that a similar impact toughness of ~100 J is obtained in the studied steel by increasing the tempering temperature to 500 °C, with a corresponding reduction in YS and UTS to 1090 MPa and 1180 MPa, respectively. These results show the significant potential of Q&P to improve YS–toughness combinations compared to conventional low-temperature tempering.

## 4. Discussion

### 4.1. Microstructural Evolution during Q&P

The obtained results suggest that the tempering of martensite and the competition between the stabilization of RA, bainite transformation, and the precipitation of carbide particles during the partitioning stage of the Q&P treatment are the key processes that determine the final microstructure and strength–toughness combinations. The continuous dilatation during the isothermal holding of the steel initially quenched to 210 °C is shown in Figure 10. The heating rate from QT to PT during dilatometric analysis was set to 100 °C, which corresponds to the experimentally measured heating rate for the steel samples in the salt bath [45]. Under these conditions, the PT is reached in about 1s; then, the steel microstructures evolve under isothermal holding. As mentioned above, the volume fraction of RA decreases as the Pt increases, and the volume fraction of bainitic ferrite also increases. It is apparent that the dilatation of the sample during the partitioning stage is mainly caused by the formation of bainitic ferrite. Nevertheless, the volume fraction of the bainitic ferrite is difficult to estimate precisely from the dilatation curve, as the volume expansion caused by the austenite decomposition is counteracted by the volume contraction due to the increase in the volume fraction of the transition carbide particles in the primary martensite during tempering [39,46]. The slope of the dilatometric curve approaches zero after 600 s at 350 °C, indicating that the bainite transformation is nearly completed within this holding time [41].

Following Equations (1) and (2), the carbon content in RA corresponding to an *Ms* temperature of 20 °C is determined to be ~1.0 wt.%. This suggests that the carbon enrichment of RA occurs within a short period and eliminates the formation of secondary martensite, even for samples subjected to short partitioning for 20 s; this finding agrees with the dilatometric data and the microstructural observations. Figure 11 shows the width distributions of the RA islands in bainitic ferrite in the steel initially quenched at 210 °C and subjected to partitioning for 20 s and 600 s. The decrease in the volume fraction of RA is mainly caused by the decomposition of coarse austenite islands. This suggests that a decrease in the amount of RA in the steel as the Pt increases is accompanied by a simultaneous growth of the transition carbide particles and the progressive redistribution of carbon in the coarse austenite islands.

To easily evaluate the partitioning of carbon from the martensite/bainitic ferrite to RA at different quenching and partitioning temperatures, kinetic simulations were performed at 210 °C and 350 °C using the DICTRA software, assuming constrained paraequilibrium conditions and a stationary interface. The martensite and austenite cell sizes were set to 100 nm and 20 nm at 210 °C and to 2.0 µm and 0.5 µm at 350 °C, respectively. The obtained carbon concentration profiles are shown in Figure 12.

According to the simulation results, carbon partitioning during holding at 210 °C for 30 s is characterized by a diffusion length of ~10 nm, which is unable to stabilize a significant amount of RA. In contrast, the diffusion length of carbon in γ-Fe reaches 0.2 µm after 20 s at a simulation temperature of 350 °C; this is comparable with the mean width of the RA islands that were observed experimentally. Further increasing the simulation time to 600 s leads to carbon content that is nearly balanced in the whole austenite cell. Several scenarios may be suggested to explain the decrease in the RA fraction as the Pt increases. First, the austenite decomposition may be related to the precipitation and growth of carbide particles in the ferrite matrix adjacent to RA. Hidalgo et al. point out that austenite regions near the martensite–austenite interface are more likely to have increased carbon concentrations, compared to those in the austenite grain interiors, and thus are prone to decomposing to ferrite and carbides [47]. Second, although the carbon concentrations in the RA in the Q&P samples measured using the XRD method are nearly the same, heterogeneities in C concentrations are possible and may result in the decomposition of RA through geometrical partitioning by the bainitic ferrite sheaves. The latter assumption is supported by recent study on carbon heterogeneities in austenite during Q&P, which used the high-energy X-ray diffraction technique [48]. The experimental observation of the preferential decomposition of the coarse austenite islands during partitioning also suggests the redistribution of carbon within sizable RA islands. The amount of the secondary martensite formed during the final quenching to room temperature in the steel after short-term partitioning for 20 s can be assumed to be negligibly small due to the large measured fraction of primary martensite and retained austenite of ≥90% and the rapid bainite transformation that is accompanied by the stabilization of RA due to carbon enrichment.

### 4.2. Relationship between Microstructure and Mechanical Properties

According to the observed variations in the tensile properties and the impact toughness of the Q&P samples, the mechanical properties of the steel are mainly affected by the Pt and less sensitive to the change in the QT. The significant increase in the yield strength with the increase in Pt can be attributed to a decrease in the amount of RA, as well as to hardening due to the precipitation of η-carbide particles in the martensite (Table 1 and Table 2). After isothermal processing at 350 °C, the steel is characterized by a high Charpy toughness value of ~100 J, suggesting that a mixture of bainitic ferrite and RA is beneficial for the material’s toughness. The relatively low strength of this microstructure as compared to Q&P may be related to the infrequent precipitation of carbide particles and the low dislocation density in bainitic ferrite (Table 2).

It is apparent that the increased strain-hardening capacity of retained austenite is responsible for the enhanced ductility of the Q&P samples with higher fraction of RA. Nevertheless, a progressive decrease in the uniform elongation and total elongation of these samples is accompanied by a significant increase in the energy absorbed during Charpy impact testing. Many researchers have reported RA to have inconsistent effects on the ductility and toughness of the steels after isothermal and Q&P treatments [17,49,50]. Zhou et al. suggested that the mechanical transformation of less stable coarse austenite islands into brittle martensite facilitates the crack initiation and fast crack propagation processes, whereas the more stable thin-film-like RA is favorable for blunting the crack tips [51]. Elsewhere, the fracture toughness significantly depended on the degree of tempering of the primary martensite matrix [27]. The load–deflection curves obtained during impact tests of the steel samples after different heat treatments are shown in Figure 13.

The peaking load of the as-quenched specimen was 11–25% higher than that of the Q&P specimens and ~40% higher than that of the specimen subjected to isothermal treatment. This finding is in good agreement with the difference in the UTS of the tensile specimens. The increased absorbed energies of the QP specimens with Pt = 600 s are mainly attributed to the significant plastic deformation prior to the fast crack propagation stage and the pronounced post-unstable ductile fracture stage. Figure 14 shows the SEM fracture surface morphology of the impact test specimens treated by Q&P. A dimple pattern resulting from ductile shearing is observed in the crack initiation zones of both the A900-Q210-P350 (20 s) and A900-Q210-P350 (600 s) specimens.

The large shear lip zones are seen on the fracture surface of the A900-Q210-P350 (600 s) specimen. A quasi-cleavage fracture morphology, composed of relatively flat facets mixed with non-uniform dimples, appears in the fast crack propagation zones (Figure 14(a2,b2)). The fast crack propagation zone of the A900-Q210-P350 (600 s) specimen contains an increased number of ductile features compared to the A900-Q210-P350 (20 s) specimen. This result indicates the A900-Q210-P350 (20 s) specimen’s lower resistance to crack propagation. To further analyze the microcrack propagation paths in this sample, the secondary cracks below the fracture surface were examined using SEM (Figure 15).

Numerous microvoids and subcracks were observed inside the initial martensite and bainite grains (Figure 15b). The microvoids are often nucleated near the RA films in the M/A constituents; meanwhile, there is no clear evidence of void or crack initiation at the grain boundaries or transition carbides. This is in accordance with the findings of Du et al., who reported that the deformation incompatibility between RA and tempered martensite leads to the preferential formation of voids and microcracks [52]. The straight propagation path of the secondary crack through the bainitic ferrite grain shown in Figure 15c also indicates that the coarse islands of RA cannot effectively inhibit crack growth. Thus, the increased impact toughness of the QP specimens with Pt = 600 s is attributed not only to the higher degree of tempering of martensite but also to the refinement of the RA islands during the partitioning stage.

## 5. Conclusions

Q&P processing was applied to low-alloy 0.25C steel to achieve multiphase microstructures composed of tempered martensite, bainite, and RA. The tensile properties and the impact toughness of the Q&P samples were investigated and compared to those obtained after water quenching and an isothermal treatment. The main results are as follow:Quenching to 210–230 °C with subsequent partitioning at 350 °C allows for the stabilizing of 6–11% of RA, and the brittle secondary martensite is eliminated during the final cooling. Stabilized RA with a film-like morphology is located on the lath and block boundaries of the tempered martensite, while relatively wide RA islands with irregular shapes are the product of the bainite transformation of large austenite regions during the partitioning stage.The bainitic transformation in the Q&P samples is nearly completed within 600 s at 350 °C. A decrease in the volume fraction and the mean width of RA as the Pt increases from 20 s to 600 s is mainly caused by the decomposition of coarse austenite islands.The concurrent increase in the yield strength and decrease in ductility with the increasing Pt is attributed to the decomposition of less strengthened RA and the precipitation of η-carbide particles in martensite.The large austenite islands have a detrimental effect on the fracture resistance of the Q&P steel. Martensite tempering and the stabilization of RA during partitioning is responsible for an excellent combination of a high-impact energy of ~100 J and a yield strength of 1200 MPa.

## Figures and Tables

**Figure 1 materials-16-03851-f001:**
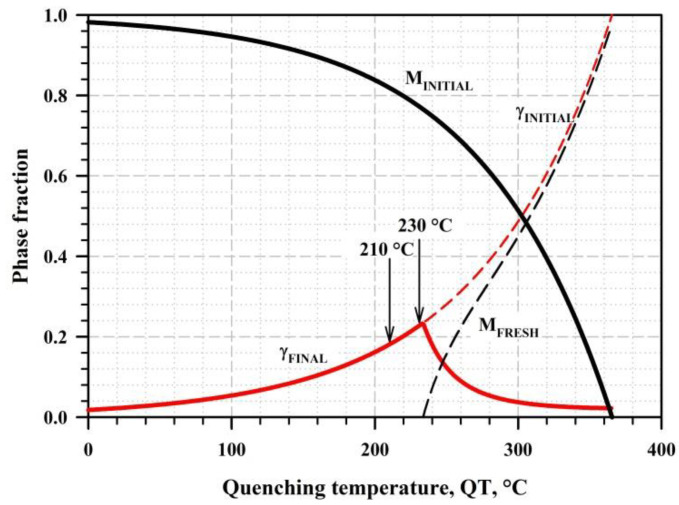
Phase fractions as functions of the quench temperature, derived using the K–M relationship.

**Figure 2 materials-16-03851-f002:**
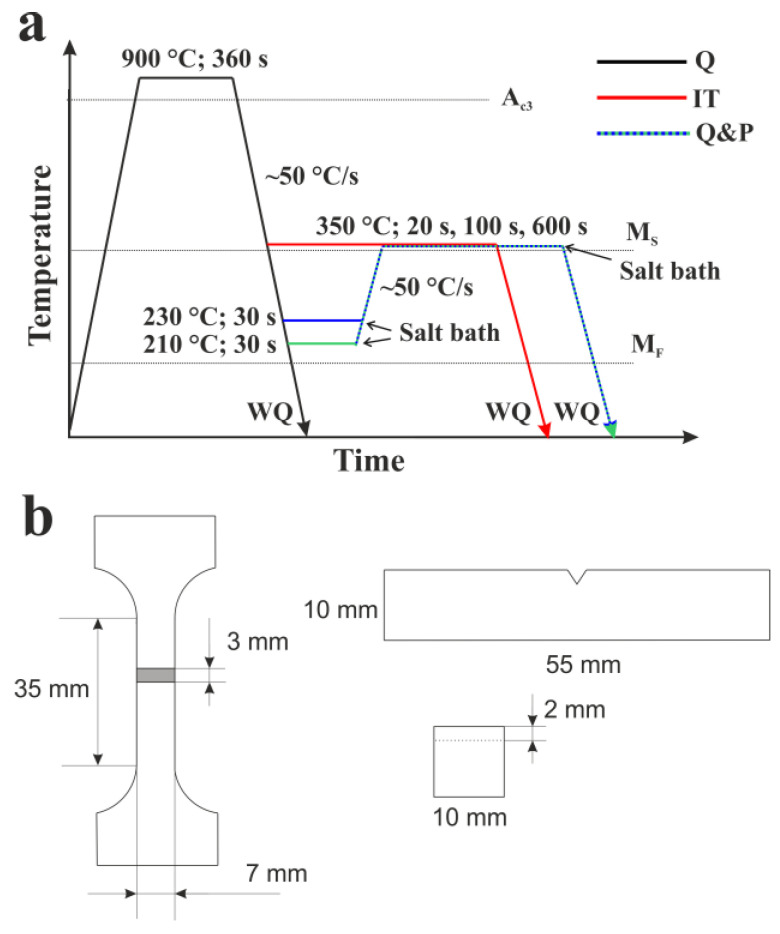
(**a**) The time–temperature regimes used for the samples subjected to water quenching (as-quenched), isothermal treatment (IT), and Q&P. (**b**) Dimensions of tensile and Charpy impact specimens.

**Figure 3 materials-16-03851-f003:**
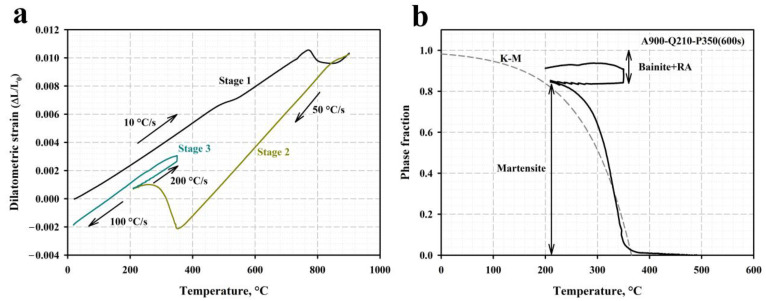
(**a**) The relative length change vs. temperature for the steel austenized at 900 °C (stage I) and quenched to 210 °C (stage II), followed by partitioning at 350 °C for 600 s with the final quenching to room temperature (Stage III); (**b**) phase fractions derived from the dilatometric analysis of the Q&P sample by applying the lever rule.

**Figure 4 materials-16-03851-f004:**
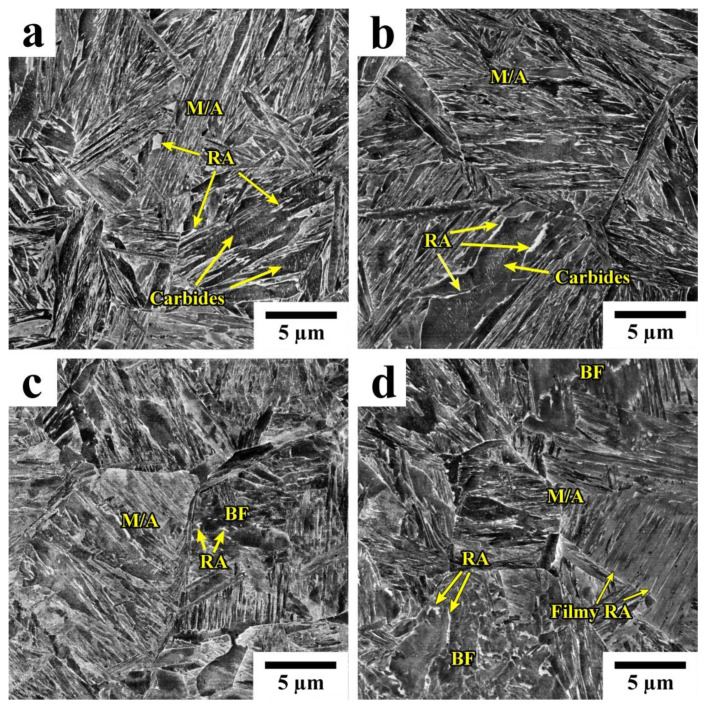
SEM microstructures of Q&P samples: (**a**) A900-Q210-P350 (20 s), (**b**) A900-Q230-P350 (20 s), (**c**) A900-Q210-P350 (600 s), (**d**) A900-Q230-P350 (600 s).

**Figure 5 materials-16-03851-f005:**
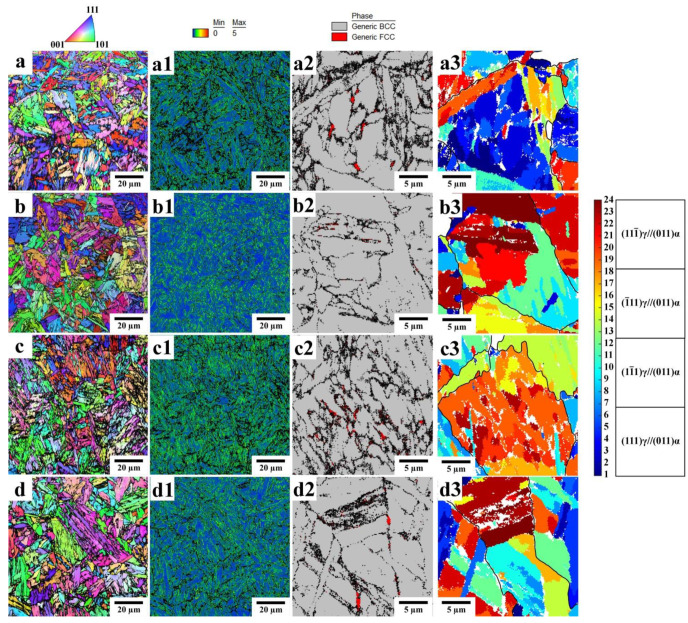
EBSD analysis of the Q&P samples: (**a**–**a3**) A900-Q210-P350 (20 s), (**b**–**b3**) A900-Q230-P350 (20 s), (**c**–**c3**) A900-Q210-P350 (600 s), (**d**–**d3**) A900-Q230-P350 (600 s). (**a**–**d**) Inverse pole figure (IPF) maps, (**a1**,**b1**,**c1**,**d1**) kernel average misorientation maps, (**a2**,**b2**,**c2**,**d2**) enlarged phase maps with RA in red, (**a3**,**b3**,**c3**,**d3**) the K–S variant maps from the enlarged regions (**a2**,**b2**,**c2**,**d2**) obtained using the parent grain reconstruction method. (In the K–S variant maps, black lines denote the reconstructed PAG boundaries).

**Figure 6 materials-16-03851-f006:**
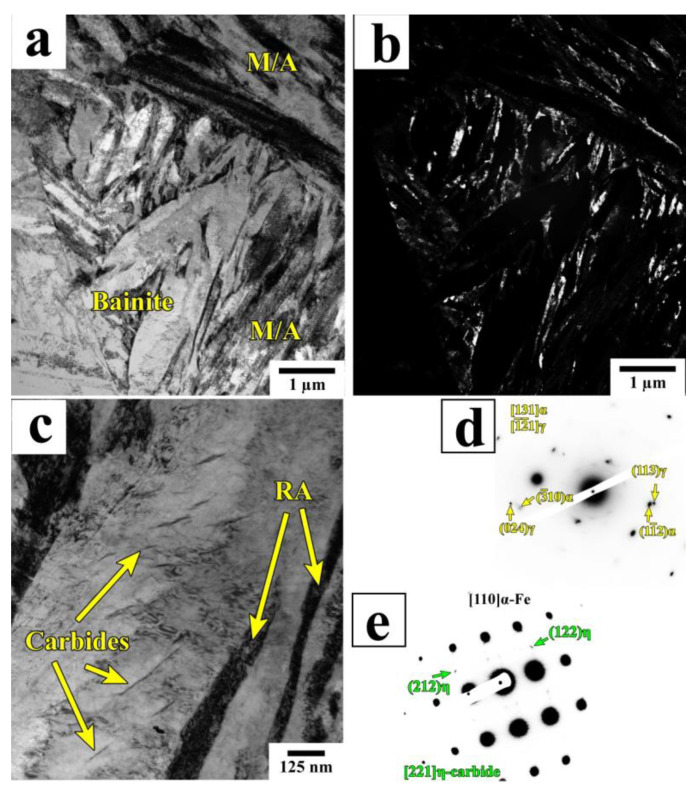
Bright-field (**a**,**c**) TEM images and (**b**) a dark-field image of RA of the A900-Q230-P350 (20 s) sample. The corresponding SAD patterns of RA and carbide particles from (**a**,**b**) and (**c**) are shown in (**d**) and (**e**), respectively.

**Figure 7 materials-16-03851-f007:**
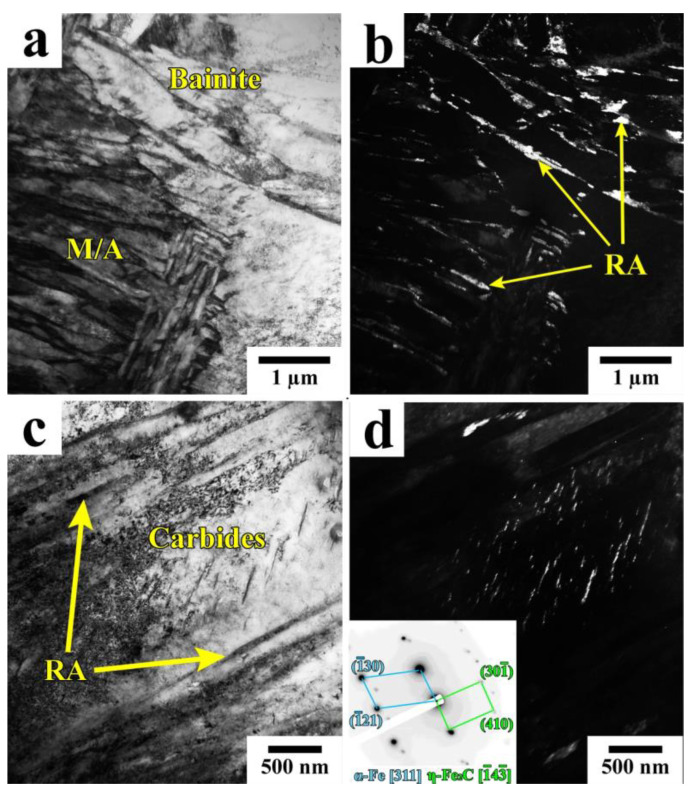
Bright-field (**a**,**c**) and dark-field (**b**,**d**) images of RA and carbide particles in the A900-Q230-P350 (600 s) sample.

**Figure 8 materials-16-03851-f008:**
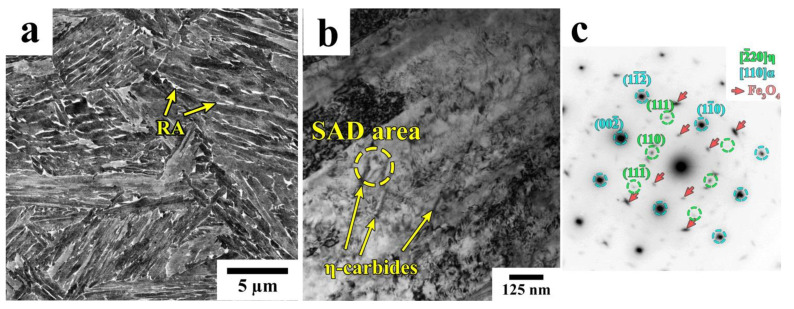
SEM (**a**) and TEM (**b**) micrographs showing the microstructure of the A900-IT350 (600 s) sample. A representative selected area diffraction pattern of the η-carbide particles is shown in (**c**). The red arrows indicate the diffraction spots from the Fe_3_O_4_ surface oxide with a zone axis of <111>.

**Figure 9 materials-16-03851-f009:**
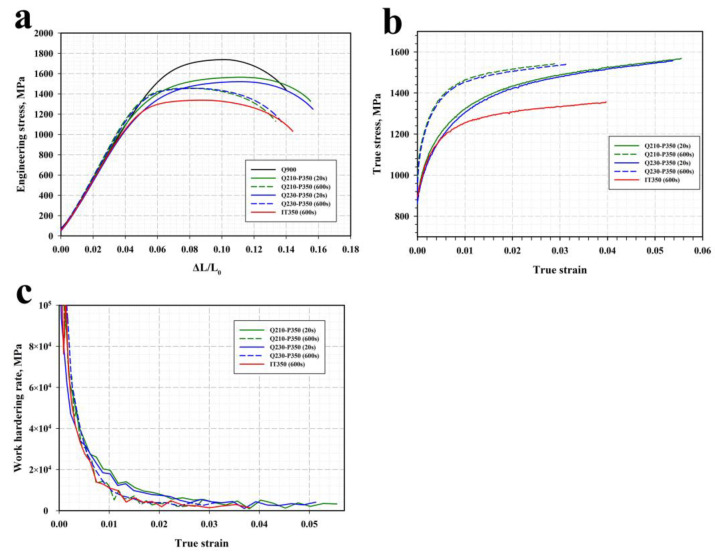
(**a**) Stress–strain curves of the steel samples subjected to different heat treatments. (**b**) True stress–strain curves of the Q&P and IT samples. (**c**) Work hardening rate as a function of true strain for the Q&P and IT samples.

**Figure 10 materials-16-03851-f010:**
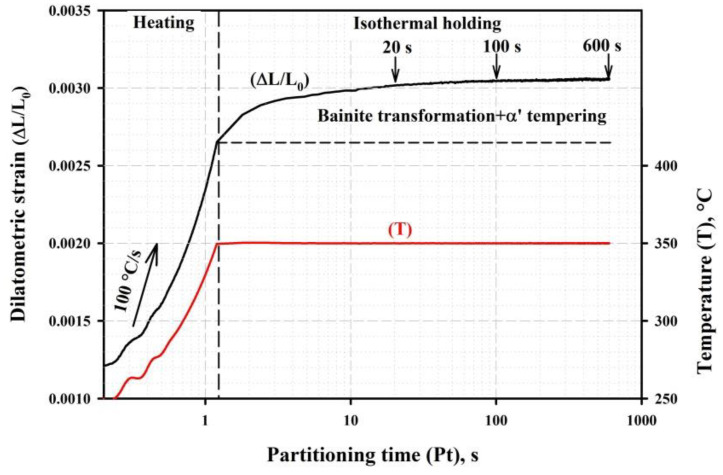
Dilatometric strain and temperature as functions of the partitioning time for the A900-Q210-P350 (600 s) sample.

**Figure 11 materials-16-03851-f011:**
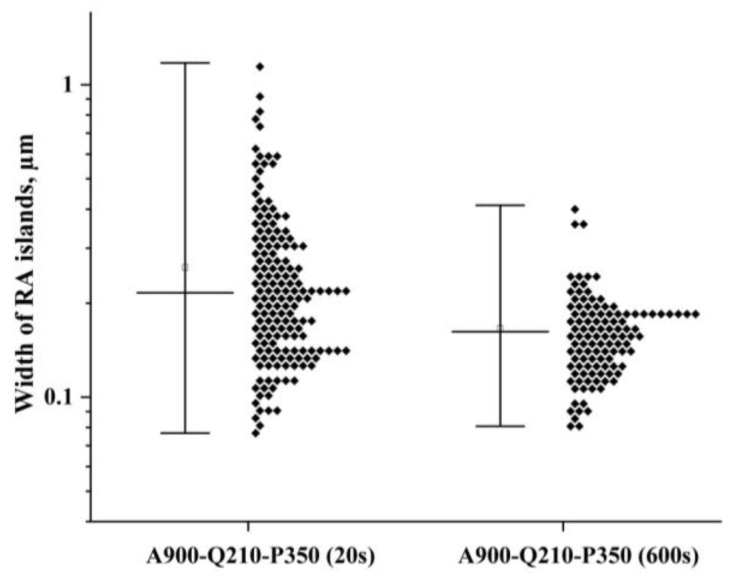
Width distributions of RA islands observed on the SEM micrographs of the A900-Q210-P350 (20 s) and A900-Q210-P350 (600 s) samples.

**Figure 12 materials-16-03851-f012:**
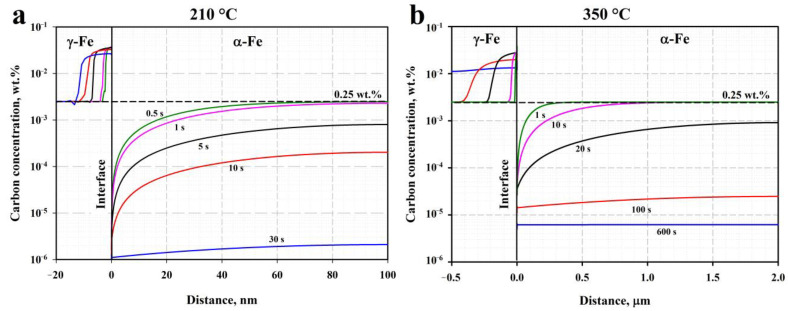
Carbon concentration profiles according to DICTRA diffusion simulations: (**a**) for partitioning times of 0.5, 1, 5, 10, and 30 s at 210 °C; (**b**) for partitioning times of 1, 10, 20, 100, and 600 s at 350 °C.

**Figure 13 materials-16-03851-f013:**
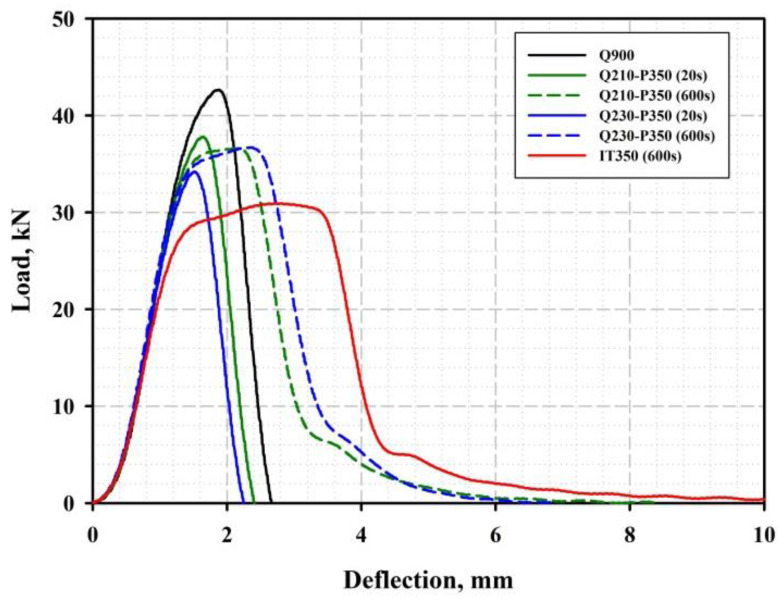
Load–deflection curves of the steel samples after different heat treatments, obtained using instrumented Charpy impact tests at room temperature.

**Figure 14 materials-16-03851-f014:**
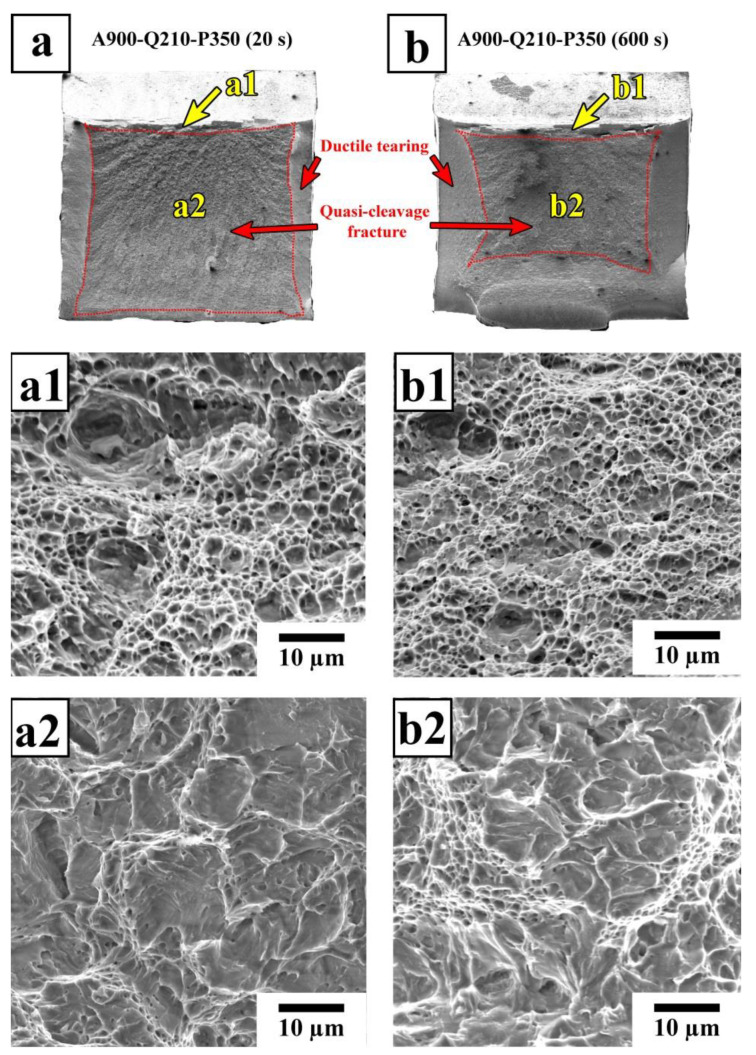
SEM fractographs of Charpy impact specimens of the steel samples subjected to Q&P: (**a**–**a2**) the A900-Q210-P350 (20 s) specimen; (**b**–**b2**) the A900-Q210-P350 (600 s) specimen. (**a**,**b**) General view of the fracture surface. (**a1**,**b1**) Fracture morphology in the crack initiation zone. (**a2**,**b2**) Fracture morphology in the fast crack propagation zone.

**Figure 15 materials-16-03851-f015:**
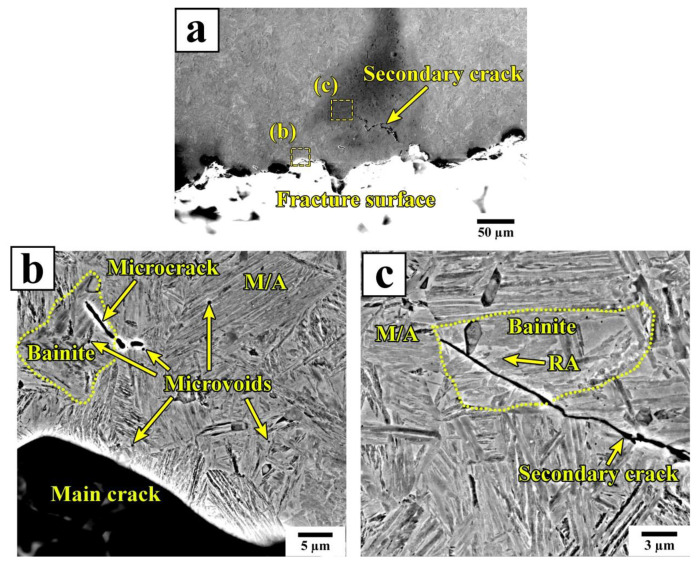
Secondary cracks and microvoids near the fracture surface in the fast crack propagation zone of the A900-Q210-P350 (20 s) specimen after impact testing. (**a**) SEM image of the cross section of Charpy specimen revealing secondary cracks near to fracture surface, (**b**,**c**) SEM micrographs of the enlarged areas highlighted in (**a**).

**Table 1 materials-16-03851-t001:** Phase volume fractions and carbon content in the retained austenite obtained using dilatometry and XRD.

Treatment	State	Phase Volume Fractions, %					Carbon in RA (XRD), wt.%
		M_INITIAL_ (Dilatometry)	RA_INITIAL_ (1 − M_INITIAL_)	RA_FINAL_ (XRD)	RA_FINAL_ (Magnetic Saturation)	Bainitic Ferrite (RA_INITIAL_ − RA_XRD_)	
As quenched	A900-Qwater	~100	-//-	-//-	1.4 ± 0.3	-//-	-//-
Q&P	A900-Q210-P350 (20 s)	85	15	10	13.3 ± 0.2	5	1.04
	A900-Q210-P350 (600 s)	85	15	7	5.3 ± 0.4	8	0.96
	A900-Q230-P350 (20 s)	83	17	11	12.3 ± 0.4	6	1.06
	A900-Q230-P350 (600 s)	83	17	6	5.2 ± 0.2	11	1.05
IT	A900-IT350 (600 s)	5	95	7	7.7 ± 0.2	88	1.07

**Table 2 materials-16-03851-t002:** The microstructural parameters of the steel samples subjected to different heat treatments.

Treatment	State	d_PAG_ (OM), µm	d_P_ (EBSD), µm	d_b_ (EBSD), µm	Length/Width of *η*-Carbide Particles, nm	KAM Average, (Scan Step is 0.15 µm), °	ρ_KAM_, m^−2^ *×* 10^15^	ρ_XRD_, m^−2^ *×* 10^15^
As quenched	A900-Qwater	16.2 ± 0.8	4.6 ± 0.4	0.89 ± 0.02	-	0.69	0.65	3.2 ± 0.5
Q&P	A900-Q210-P350 (20 s)		6.0 ± 0.8	0.79 ± 0.03	93/13	0.81	0.75	4.0 ± 0.7
	A900-Q210-P350 (600 s)		5.5 ± 0.6	0.80 ± 0.04	131/13	0.56	0.53	1.6 ± 0.3
	A900-Q230-P350 (20 s)		7.0 ± 1.1	0.95 ± 0.06	87/9	0.81	0.75	3.4 ± 0.7
	A900-Q230-P350 (600 s)		6.3 ± 1.1	0.93 ± 0.06	121/13	0.62	0.59	2.4 ± 0.4
IT	A900-IT350 (600 s)		6.7 ± 0.8	0.87 ± 0.04	126/17	0.54	0.51	1.9 ± 0.3

**Table 3 materials-16-03851-t003:** Mechanical properties of the steel samples subjected to different heat treatments.

Treatment	State	Hardness, HRC	Yield Strength (YS), MPa	Ultimate Tensile Strength (UTS), MPa	UTS/YS	Total Elongation (TE), %	Uniform Elongation, %	Charpy Toughness at 20 °C, J
As quenched	A900-Qwater	51.3 ± 0.6	1360	1740	1.28	8.8	3.6	49
Q&P	A900-Q210-P350 (20 s)	47.6 ± 1.0	1150	1560	1.36	11.0	5.6	38
	A900-Q210-P350 (100 s)	45.5 ± 0.9	1220	1440	1.18	10.2	3.9	93
	A900-Q210-P350 (600 s)	45.6 ± 0.4	1230	1450	1.18	9.5	2.9	93
	A900-Q230-P350 (20 s)	46.5 ± 1.1	1090	1520	1.39	11.2	5.5	44
	A900-Q230-P350 (100 s)	46.1 ± 0.2	1210	1460	1.21	10.5	4.0	64
	A900-Q230-P350 (600 s)	46.4 ± 0.4	1230	1450	1.18	9.9	3.3	101
IT	A900-IT350 (600 s)	43.8 ± 1.1	1070	1340	1.25	11.1	3.9	99

## Data Availability

Data will be made available on request.

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
