# Peer review of "Strength–Toughness of a Low-Alloy 0.25C Steel Treated by Q&P Processing"

_materials, 2023, doi:10.3390/ma16103851_

Round 1
Reviewer 1 Report
This is a good and interesting paper, and the presented results might broaden our understanding of the studied field. As such, the reviewer recommends publication of this work after a minor revision:
(1) The abstract should be revised to become more quantitative.
Author Response
Q: This is a good and interesting paper, and the presented results might broaden our understanding of the studied field. As such, the reviewer recommends publication of this work after a minor revision:
(1) The abstract should be revised to become more quantitative.
A: Thank you for your positive evaluation of our manuscript.
The mechanical properties after Q&P processing were incorporated in the revised abstract as follows.
“The best combinations of the yield strength of above 1200 MPa and impact toughness of about 100 J were obtained in the steel samples quenched to 210-230 °С and subjected to partitioning at 350 °C for 100-600 s.”
Reviewer 2 Report
1/ The novelty of the work must be more clearly emphasized. I can not agree that „However, the research on the improvement of strength-toughness combinations in Q&P steels with multiphase microstructures is still limited.” Actually, there are a lot of published works on this topic.
2/ „The steel plate was reheated to 1000 °C for 2 h and then hot-rolled to 12 mm thickness (70% total reduction), through 6 passes in the temperature range from 900 °C to 850 °C followed by air cooling.” It is not possible to conduct 6 passes in such a narrow temperature range. The explanation is reguired. A type of the rolling machine and corresponding parameters should be provided.
3/ Figure 2a. Heating and cooling rates must be provided for this figure to be reproduce the tests. Explain what is the effect of cooling rate from the austenitizing temperature to quenching temperature on the formation of martensite ? Moreover, explanation is required concerning the effect of heating rate from the quenching temperature to partitioning temperature. Does it affect the kinetics of possible bainite formation during the partitioning step ? What is the effect of cooling rate from the partitioning temperature to room temperature ? Does it affect the final stability of austenite and formation of fresh / untempered martensite ?
4/ Is it beneficial that bainitic ferrite is formed or not ? Are you able to predict if the carbon enrichment in retained austenite is from the martensite or from the bainitic ferrite.
6/ What is the original pgase , where carbides are formed: bainitic ferrite or martensite ? Does the martensite tempering take place during the partitioning process ?
7/ Figure 9a indicates that the tesile tests were performed wihout an extensometer ? How does it affect the measurement of elongation values ?
8/ What is the most important issue for inhibiting fresh martensite formation ?
Enslish must be improved throughout the whole manuscript
Author Response
Q: 1/ The novelty of the work must be more clearly emphasized. I can not agree that „However, the research on the improvement of strength-toughness combinations in Q&P steels with multiphase microstructures is still limited.” Actually, there are a lot of published works on this topic.
A: To clarify the novelty of the work we revised the mentioned introduction part. First, the problem was outlined in general as follows:
“While the Q&P process is widely used to improve the plasticity of cold-rolled sheet steels for automotive industry [11-15], the relationship between the strength and the impact toughness in thick plate Q&P steels with multiphase microstructures is still debatable.”
Then, the current state of research was summarized as follows.
“…Recent studies suggest that the steel microstructures composed of carbon-depleted martensite and increased fraction of RA are generally desired to accommodate a large amount of plastic strain during tensile deformation due to the transformation induced plasticity (TRIP) effect [14-16]. On the other hand, although the coarse blocky austenite significantly improves the steel ductility, it was shown that it has detrimental effect on the impact toughness [17,18]. This may be due to an increase in the effective grain size controlling the toughness and lower stability of RA which transformed to brittle sec-ondary martensite during final cooling [15,19]. Bagliani et al. argued that the presence of UM in the microstructure can have prominent effect on the mechanical behavior, especially, toughness [20]. Large UM–A constituent cannot effectively inhibit the crack propagation and results in low fracture toughness [20-22]. To improve the toughness, the bainitic transformation of untransformed islands of RA during partitioning can be utilized. The finely dispersed mixture of lath bainite ferrite, lath martensite and film-like retained austenite can effectively inhibit the crack propagation [21,22]. Huang et al. showed that the microstructure composed of martensite and bainite obtained during uncompleted isothermal transformation followed by Q&P can improve the impact toughness without significant loss in the strength [23]. “
Finally, the aim of the present study was introduced.
“… Thus, the strength – toughness relationship in Q&P steels deserves more detailed investigation. The study on the evolution of retained austenite during bainite transformation at the partitioning stage and its effect on the impact behavior is of particular interest.”
Q: 2/ „The steel plate was reheated to 1000 °C for 2 h and then hot-rolled to 12 mm thickness (70% total reduction), through 6 passes in the temperature range from 900 °C to 850 °C followed by air cooling.” It is not possible to conduct 6 passes in such a narrow temperature range. The explanation is reguired. A type of the rolling machine and corresponding parameters should be provided.
A: The parameters of the hot rolling procedure were clarified in the Materials and Methods section of the revised manuscript as follows.
“To maintain thermal rolling condition the plate was reheated to 900 °C for 5 min between each pass. A two-roll mill with a maximum rolling force of 2500 kN was used.”
Q: 3/ Figure 2a. Heating and cooling rates must be provided for this figure to be reproduce the tests.
A: The heating and cooling conditions were provided in the revised Figure 2a.
Q: Explain what is the effect of cooling rate from the austenitizing temperature to quenching temperature on the formation of martensite?
A: The effect of the cooling rate during first step of Q&P processing was clarified in the revised manuscript as follows.
“The cooling rate of the steel samples in the salt bath is about 50 °C/s, that is well above the critical cooling rate and thus sufficient to form the initial martensite during the first step of Q&P processing [20].”
Q: Moreover, explanation is required concerning the effect of heating rate from the quenching temperature to partitioning temperature. Does it affect the kinetics of possible bainite formation during the partitioning step?
A: The effect of heating rate on the bainite formation was discussed as follows.
“The heating rate from QT to PT during dilatometric analysis was set as 100 °C that corresponds to the experimentally measured heating rate for the steel samples in the salt bath [46]. Under this condition the PT is reached in about 1s and then the steel microstructures evolve under isothermal holding. As mentioned above, the volume fraction of RA decreases with increasing Pt, and the volume fraction of bainitic ferrite increases. It is apparent that the dilatation of the sample at the partitioning stage is mainly caused by the formation of bainitic ferrite. Nevertheless, the volume fraction of the bainitic ferrite is difficult to estimate precisely from the dilatation curve as the volume expansion due to the austenite decomposition is counteracted by the volume contraction due to increase in the volume fraction of the transition carbide particles in the primary martensite during tempering [40,47].”
Q: What is the effect of cooling rate from the partitioning temperature to room temperature ? Does it affect the final stability of austenite and formation of fresh / untempered martensite ?
A: In the present study the absence of length increase during the final quenching to room temperature after partitioning at 350 °C for 600 s implies that no secondary martensite was formed upon cooling to room temperature. The corresponding discussion was amended as follows.
“The possible amount of the secondary martensite formed during final quenching to room temperature in the steel after short-term partitioning for 20 s can be assumed to be negligibly small due to the large measured fraction of primary martensite and retained austenite of ≥90 % and the rapid bainite transformation that is accompanied by stabilization of RA by carbon enrichment.”
Q: 4/ Is it beneficial that bainitic ferrite is formed or not? Are you able to predict if the carbon enrichment in retained austenite is from the martensite or from the bainitic ferrite.
A: The discussion on the microstructure – mechanical properties relationship was expanded to clarify the effect of bainitic ferrite on the strength and toughness of the steel.
“The steel after isothermal processing at 350 °C is characterized by the high value of Charpy toughness of ~100 J suggesting that mixture of bainitic ferrite and RA is beneficial for the toughness. The relatively low strength of this microstructure as compared to Q&P can be related to the rare precipitation of carbide particles and the low dislocation density in bainitic ferrite (Table 2).”
We will consider the carbon enrichment process during partitioning stage in future works.
Q: 6/ What is the original pgase , where carbides are formed: bainitic ferrite or martensite ? Does the martensite tempering take place during the partitioning process ?
A: The microstructure characterization shows that the major part of the η-carbides precipitates in the coarse laths of primary martensite during partitioning stage as mentioned in Section 3.1 of the revised manuscript.
In addition, small amount of carbides precipitates during transformation of RA to bainite. The detailed analysis on the competition between stabilization of RA, bainite transformation and precipitation of carbide particles is planned to be done in future work.
Q: 7/ Figure 9a indicates that the tesile tests were performed wihout an extensometer ? How does it affect the measurement of elongation values ?
A: The details of the tensile tests were added to the revised version of manuscript as follows.
“The elongation and reduction of cross section area of tensile specimens were recorded by non-contact digital image correlation (DIC) measurement system.”
Q: 8/ What is the most important issue for inhibiting fresh martensite formation ?
A: The carbon enrichment of RA shifts the Ms temperature to subzero temperatures that inhibits the formation of fresh martensite as it was predicted by Eq. (1) and Eq. (2). (Figure 1).
Based on the present results it can be concluded that the large measured fraction of primary martensite and retained austenite of ≥90 % and the rapid bainitic transformation that is accompanied by stabilization of RA by carbon enrichment inhibits the fresh martensite formation.
Q: Enslish must be improved throughout the whole manuscript
A: The manuscript has been revised carefully to enhance the quality of presentation.
Reviewer 3 Report
Strength - Toughness improvement of all structural materials especially steels are of great importance to the society. With in-depth analysis of the background, the authors have performed sufficient work to optimize the Q&P processing approaches, investigate the microstructure, measure the mechanical properties and most importantly analysis the mechanism of the improvement. In my opinion, this is an excellent work, I have no comments in the technical regard. Here, I would like to felicitate the authors on this competently executed study and gladly suggest acceptance without any revision.
The quality of English Language is high.
Author Response
Thank you for your positive evaluation of our manuscript.
Reviewer 4 Report
The authors provided colorful Q&P processes and IT process to control the microstructure of the 0.25C low alloyed steel. The microstructure evolution was systematically investigated and discussed. This work is valuable for the development of advanced high strength steels.
1. In the present work, a 12 mm hot-rolled plate was fabricated. The Q&P process were widely used to improve the plasticity of cold-rolled advanced high strength steels. Why is a heavy plate used?
2. Why is the microstructure of the IT sample not provided? SEM, EBSD?
Author Response
Thank you for your positive evaluation of our manuscript.
Q: 1. In the present work, a 12 mm hot-rolled plate was fabricated. The Q&P process were widely used to improve the plasticity of cold-rolled advanced high strength steels. Why is a heavy plate used?
A: The thick hot-rolled plate in the present study was fabricated to investigate the effect of Q&P treatment on the CVN impact energy in accordance with the ASTM E-23 standard on the 10x10x55 mm specimens.
Q: 2. Why is the microstructure of the IT sample not provided? SEM, EBSD?
A: The SEM ant TEM micrographs of the IT sample were presented in Fig.8 and the microstructural parameters of IT sample including the data obtained from EBSD analysis were summarized in Table 2.
Reviewer 5 Report
To be honest, I wondered to make decision, because the report itself was described well; I thought microstructural characterization was excellent. However, the current developing target of advanced steel is the tensile strength of 1.5 GPa and total elongation of 20% or so, for examples, see https://isma.jp/steelsheet/ (English version is available)" , which has already been achieved by steel companies. Hence, the elongation of approximately 5% in this paper is not attractive at all. Therefore, it was difficult for me how to advice to improve the present manuscript to meet the current demand. I have to say that the novelty of this study itself was poor (but the quality of experiments and discussion are good). Incidentally, a minor comment includes that "engineering strain" in Fig. 9a must be incorrect: it seems to be the crosshead displacement divided by the specimen gauge length.
Author Response
Thank you for your positive evaluation of our manuscript.
In the introduction section of the revised manuscript we emphasized the novelty of the present study that is being carried out to improve the strength-toughness combinations in thick plate Q&P steel.
In the present study we showed that it is possible to achieve relatively high values of YS, UTS and CVN impact energy, concurrently. Total elongation of 10% was also attained (see Table 3). The hot-rolled plates from the steel with such combination of mechanical properties will meet demand for commercial use, for instance, in agriculture machinery. We also investigate the Q&P behavior of the similar steel with 0.4C (https://doi.org/10.3390/met13040689). However, this steel exhibits relatively low CVN energy <25 J. We will discuss fracture toughness of Q&P steels in future works.
Q: Incidentally, a minor comment includes that "engineering strain" in Fig. 9a must be incorrect: it seems to be the crosshead displacement divided by the specimen gauge length.
A: Corrected in the revised manuscript.
Round 2
Reviewer 2 Report
The authors addressed my comments and implemented the required changes to the current version of the manuscript
Moderate revision required
Reviewer 5 Report
The paper was improved taking the reviewer's comments into consideration.
English was understandable.